# ANAVI: Audio Noise Awareness using Visuals of Indoor environments for NAVIgation

**Vidhi Jain**          **Rishi Veerapaneni**          **Yonatan Bisk**

Carnegie Mellon University

**Abstract:** We propose Audio Noise Awareness using Visuals of Indoors for NAV-Igation (ANAVI[1]) for quieter robot path planning. While humans are naturally aware of the noise they make and its impact on those around them, robots currently lack this awareness. A key challenge in achieving audio awareness for robots is estimating *how loud will the robot's actions be at a listener's location?* Since sound depends upon the geometry and material composition of rooms, we train the robot to passively perceive loudness using visual observations of indoor environments. To this end, we generate data on how loud an 'impulse' sounds at different listener locations in simulated homes, and train our Acoustic Noise Predictor (ANP). Next, we collect acoustic profiles corresponding to different actions for navigation. Unifying ANP with action acoustics, we demonstrate experiments with wheeled (Hello Robot Stretch) and legged (Unitree Go2) robots so that these robots adhere to the noise constraints of the environment.

**Keywords:** Robots, Acoustic Noise, Vision, Learning

## 1 Introduction

Humans are very aware of the noise that their movements create. In homes and offices, we avoid being noticed if a person is having a video call or a child who has just fallen asleep. Home robots need that same social awareness when planning their actions in indoor environments. Unfortunately today robot vacuums may be as loud as 70 decibels, which concerns many people with sensitive ears. The existing solution is to use the "do not disturb" mode [1] which completely stops the robot from functioning so that it does not make noise. Integrating more complex robots, e.g., quadrupeds, in homes, needs more sophisticated methods of addressing noise levels while maintaining efficiency.

Although sound is inevitable with robot movement, it can be mitigated. In isolation, e.g. without acoustic reflections or echoes, unoccluded sound intensity decays quadratically with distance. As a robot moves away from humans or sound-sensitive areas, its sounds, such as motor noise or beeps, become less likely to be heard. Thus, a simplistic approach to reducing this noise is for the robot to move slowly and steer clear of people and pets in its environment. While this might initially seem like a sensible strategy, it can lead to increased time and energy consumption for task completion and may not always be practical or feasible. Importantly, sound intensity depends on many other factors beyond distance. The intensity at the same distance of 1 meter will be reduced if separated by a wall or in a carpeted room, whereas, conversely, halls create echoes. Architectural geometry and material properties affect how the sound reflects, diffracts, and is absorbed.

Existing work on audio for robotics focuses on finding the source of an audio signal, guiding navigation [2, 3]; which leverages realistic 3D simulators to learn how to navigate to the sound of dripping water in the sink or a fire alarm. In contrast, we focus on awareness of self-generated sounds; a related yet orthogonal problem: how loud will the robot's actions be at a listener's location?

To this end, we propose Audio Noise Awareness by Visual Interaction (ANAVI), a framework that allows a robot to learn about the unintended noise made by its actions and thereby adapt its actions to the noise constraints of the environment. We train an Acoustic Noise Prediction (ANP) model using

---

[1] "anavi" also implies peace-loving, kind to people.

8th Conference on Robot Learning (CoRL 2024), Munich, Germany.

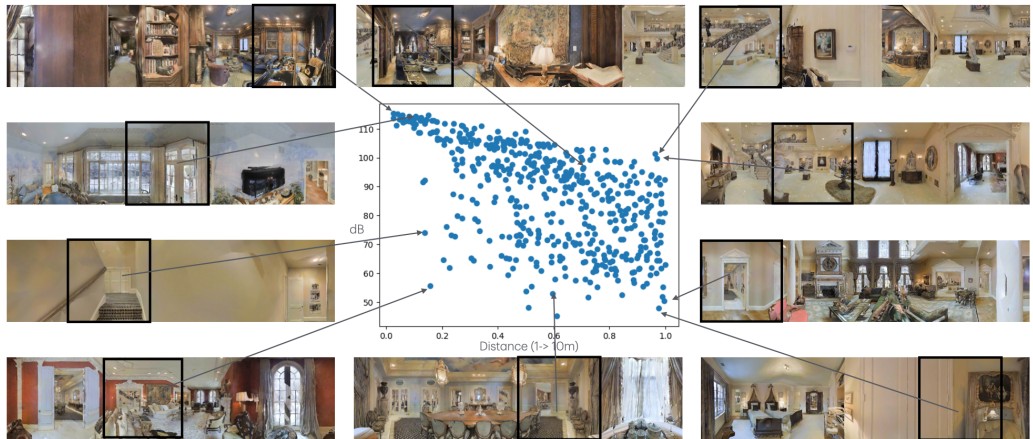

Figure 1: We generate data for acoustic impulse response in simulated 3D scenes from real-world environment scans. x-axis shows the relative distance of the listener from the source agent, y-axis shows the max decibels of a simulated Impulse Response (IR) at listener. The box area on the panaroma shows the listener's direction relative to the agent. Note the complex response pattern as materials, objects, and geometry have non-linear interactions with the sound.

audio impulse response simulation (SoundSpaces) in 3D real-world scans (Matterport). Intuitively, ANP learns to predict how an impulse generated at the source (robot's location) will be heard at a listener's location, using the relative distance and direction of the listener and, more importantly, using the visual features around the robot. ANAVI framework makes use of ANP model for estimating audio noise costs for planning a quieter robot path.

The contributions of our work are as follows. (a) Our Acoustic Noise Prediction (ANP) model with visual features performs better than distance-based heuristics and learned baselines. We show that ANP predictions overlap well with the data distribution and have higher epsilon thresholded accuracy (a metric analogous to PR curves) than distance-based baselines. (b) We show qualitative analysis of the ANP model's prediction for fixed robot location (while assuming the listener at every point on the map) and fixed listener locations (while assuming the robot at every point on the map). (c) Further, we perform real-world experiments to compare real audio decibels with our model's predictions and propose ANAVI framework for adapting robot's navigation plans for quieter paths.

## 2 Acoustic Noise Prediction (ANP)

A robot navigating indoor environments produces sound, either from its motors or its speakers trying to communicate with others in its environment. Given the robot in an environment and a listener elsewhere, we want to predict how loud the robot is at the listener's location. Sound travels from the robot's location to a listener's location depending on the relative distance, architectural geometry, materials, and hearing sensitivity of the listener. In our simulated environments, the most dramatic differences happen at short distances blocked by walls or furniture, versus long echoing hallways.

**Simulator Setup.** We create a simulated dataset of sound decibels perceived at the listener's location when the robot is located in different parts of home environments. We use the Matterport3D dataset, which consists of 85 real-world scans of indoor environments. The simulator assumes a sound impulse at the source location and calculates a response (aka Room Impulse Response or RIR) at the listener location. For this, we use Sound-Spaces 2.0 [4] to simulate how audio waves travel from the source to arrive at the listener as an impulse response. To simulate any sound, the room impulse response is convolved with the audio at the source. As convolution is linear, the max decibel of audio at the listener is linearly proportional to the decay of the max db of the RIR from that of the original impulse. We assume that an impulse generates the sound pressure level of $1W/m^2$ at the source and that the listener is within 10 m of the source. More details are in Appendix B.2.

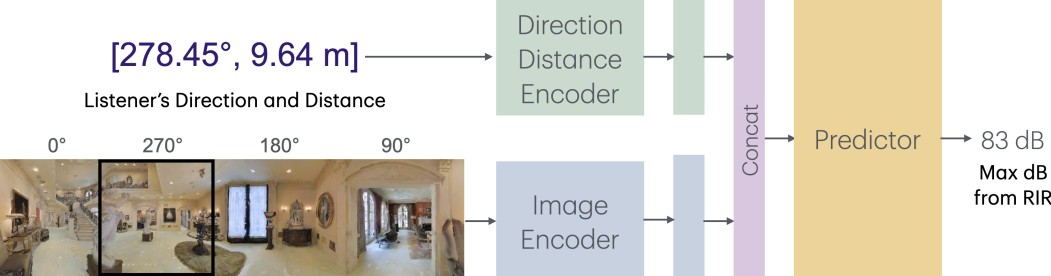

[278.45°, 9.64 m]

Listener's Direction and Distance

Direction Distance Encoder

0°   270°   180°   90°

Concat

Predictor → 83 dB

Max dB from RIR

Image Encoder

RGB Panorama from Sound Source (Robot)

Figure 3: **Architecture.** Our Acoustic Noise Prediction (ANP) model consists of *Image encoder*, *Direction-Distance encoder* and *Predictor* modules. The inputs are the 360°RGB panorama view at the robot's location, and the relative polar coordinates of the listener. The square box on the panorama highlights robot's current facing direction, and is drawn for illustration purposes only. The output is the max decibel (dB) of the Room Impulse Response (RIR) at the listener's location.

**Data.** We approximate the max dB of an audio at listener = (max dB of the RIR / max dB of the impulse at source) * max dB of audio source. Sound decibels heard by the listener as an impulse response generated at the robot's location vary as shown in Fig. 2. Note the two modes in the distribution at 68dB and 110 dB. From each map, we sample a dataset of $\mathcal{D} = \{s_i, l_i\}_{i=1}^{N}$, where $N$ is the number of samples. For each pair of robot and listener locations $\{s_i, l_i\}$, we record the visual observation at the robot $V_s$, relative polar coordinates of the listener $\{r_{sl}, \theta_{sl}\}$ and the max decibels of the impulse response generated at the listener's location $y$. The relative distance $r_{sl}$ is an important feature in predicting sound intensity heard by the listener based on spherical spreading of sound

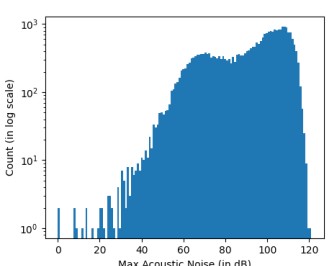

Figure 2: Histogram of max decibel values from simulated data

waves. To render realistic sounds, the simulator uses additional privileged information about 3D mesh geometry and material sound properties for ray tracing [5].

Visual observations of the surroundings contain partial information, and we aim to train a model that learns to predict the audio at the listener based on the latent visual features. Recall our example in Sec. 1, where a 1m distance in the same room versus separated by a wall affects loudness at the listener's locations. A robot should learn to perceive visual features like a wall, open area, corridors, etc., to make an informed estimate about the perceived loudness at the listener's location. For panoramic visual input, the direction of the sound-sensitive listener's location $\theta_{sl}$ is important to infer what geometry and materials are in the area the sound waves will most likely pass through on their way to the listener. We also consider ego-centric visual features with robot facing in the direction of the listener. More details comparing the ego- and pano- visual features in Sec. 3.2.

**Architecture** We learn an acoustic noise predictor function $f$ such that $y = f(r_{sl}, \theta_{sl}, V_s)$. Fig. 3 shows our proposed architecture for Acoustic Noise Prediction (ANP). Our model consists of three components. *Image Encoder* extracts the visual features using pretrained visual encoders like ResNet-18. We apply adaptive average pooling and normalize it to get visual encoding $e_{visual}$. *Direction-Distance Encoder* is a simple linear projection to obtain an encoding $e_{dirdis}$. *Predictor* takes the concatenated vector of $e_{visual}$ and $e_{dirdis}$, and processes it through a series of blocks. Each block contains a linear layer, a batch norm layer, and GeLU activation. The output size of each successive linear layer shrinks to act as an "information bottleneck". The final output is either a scalar real number for regression or $m-$ binned logits for classification.

**Training** Our code uses PyTorch and torchvision ResNet-18 checkpoint. We apply ResNet-18 pre-processing on $V_s$ for mean 0 and std 1. We normalize the $r, \theta$ and $y$ between 0 and 1. We use Huber Loss[6] with $\delta = 0.1$ and the AdamW optimizer, which has a learning rate of 0.01 and a step schedule with 0.95 decay after 10 epochs. We use a batch size of 64 on a 24GB RAM GPU.

# 3 Simulation Experiments

We assess our acoustic noise prediction model in simulation and demonstrate its applicability to real-world environments. Our simulation experiments are conducted in the Habitat 2.0 simulator [7] on the Matterport3d dataset [8]. We inherit the train/val/test splits for Matterport3d scenes from the PanoIR [4] dataset - 59 maps for train, 11 for val, and 15 for test. For training, we create 5,000 samples per map, where each sample consists of robot location $s$, listener location $l$, the robot's panoramic view $V_s$, and the listener's Impulse Response $w_l$. We post-process the impulse response to calculate the max dB heard at the listener's location as $y$. For val and test, we create 500 samples per map. We evaluate $500 \times 15 = 7500$ samples in all simulation experiments.

**Metrics.** We compare acoustic noise prediction models with data distribution coverage plots. To understand the coverage of predicted values with respect to the true labels, we include a distance-decibel plot in Fig.4. The higher the overlap of the red overlay over the blue, the better the model's performance. Note, the highly diverse data distribution of decibels $y$ for a given distance $r$.

Another, more quantitative, view of the regression analysis is via $\epsilon$-thresholded accuracy curves. Our $\epsilon$-thresholded accuracy curves are inspired by Precision-Recall curves used in classification. This comparison allows us to see the effect of cross-entropy style training, both on performance and how it skews the data distribution. See Fig. 5. Let $y$ be the normalized max decibel value of the simulated impulse response (IR)s, and $\hat{y}$ be the predicted value by the model. Let there be $N$ samples, $\mathbf{1}(\cdot)$ is the indicator function, which is 1 if the condition inside is true and 0 otherwise, and $N = 7500$. For $\epsilon$-accuracy, we compute $\frac{1}{N} \sum_{i=1}^{N} \mathbf{1}(|y_i - \hat{y}_i| \leq \epsilon)$. An $\epsilon = 1/128 = 0.007$ captures whether the model can learn to accurately predict within a single decibel. Humans typically can barely differentiate the 1 dB difference for sounds in 1000-5000 Hz frequencies, also known as Just Noticeable Difference (JND) [9]. JND increases to 3-5 dB at very low or very high frequencies and changes with larger initial loudness levels.

**Baselines.** We compare the performance of our Acoustic Noise Prediction (ANP) model with a distance-based heuristic and two learned baselines: (i) *Heuristic Distance based (Heuristic)* Sound intensity decays by a factor of inverse squared distance, that is $\frac{1}{r^2}$. Thus, the predicted max dB IR can be calculated by $-20 \log_{10}(r) + 120$. (ii) *Regression with Distance (DisLinReg)* We train a linear regression model to match the average shape and curvature of the data as a function of the data. Note that this more accurately captures the shifting mass distribution than the heuristic; however, it is unimodal in its predictions. (iii) *MLP Regression with Distance and Direction (DirDisMLP)* We train a multi-layer perceptron model with blocks of linear layers, batch norm, and ReLU activation, where layer sizes of 2, 8, 8, and 1. (iv) *Visual features with Distance and Direction (VisDirDis)* One concern with the DirDisMLP is that it does not incorporate visual information. Therefore, it cannot distinguish between scenarios with a wall between the robot and the listener versus instances in open spaces where sound may travel collision-free. In VisDirDis model, we use a panoramic RGB view at the agent's location as the visual feature of 512-dim. We include direction as input to localize and attend to how the loud listener hears the robot. The direction and distance are encoded by a linear projection to 16-dim. We concatenate and pass the resulting 528-dim vector to the predictor. The predictor consists of 4 blocks with layers sizes: 528, 256, 64, 8 and outputs as a scalar value.

## 3.1 Results

We show significant performance improvement by incorporating visual information. Visual information conveys the architectural geometry and materials used in the space, which affects the perceived sound intensity at the listener's location. Fig. 4 shows the predicted data distribution (in red), overlayed on the true test data (in blue). The ideal model would directly match the blue data distribution, and it is apparent that our model most closely aligns with the ground truth. Additionally, Figure 5 shows $\epsilon-$accuracy for learned models. On the far left of Figure 4, the geometry spreading-based distance heuristic (Heuristic) only predicts correctly for free space and doesn't account for reflections in corridors or absorption by walls in home environments. Second, we plot the linear regression prediction (DisLinReg) based on distance. While it improves the overall $\epsilon-$accuracy curve in Fig. 5, it doesn't cover the plot. The learned MLP (DirDisMLP) slightly improves performance and begins

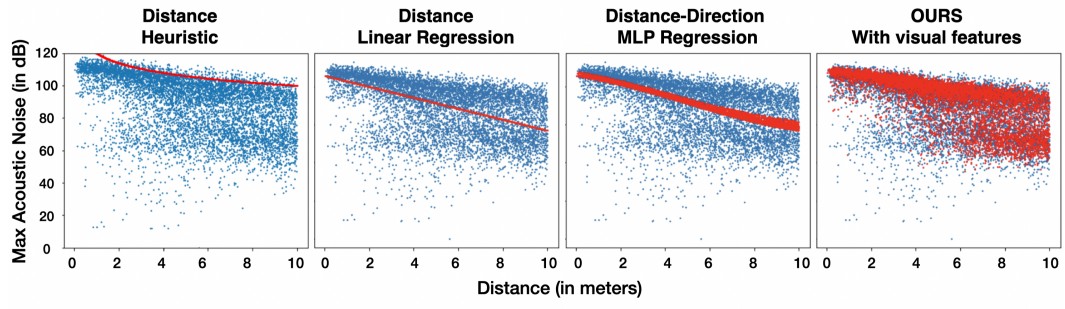

Figure 4: **Main Results.** Predicted Data Distribution plots for (left to right) Heuristic, DisLinReg, DirDisMLP, **Ours**(Pano)-VisDirDis.

to spread out the generated data distribution. But it lacks sufficient signal about the spatial context that could inform if there are possible reflections, absorptions, and diffractions of the sound. These observations validate our approach on the right, where we use visual features along with distance and direction (Pano-VisDirDis) and observe the best $\epsilon-$accuracy and data distribution coverage.

## 3.2 Ablations

We evaluate modeling design choices for loss functions and visual representations. Refer to Fig. 6 for the data distribution coverage results for all our model variants in this section. (i) *Loss function: Regression vs Classification.* We use $m-$bins to discretize the prediction and convert it from regression into a classification problem—the granularity of the prediction changes depending on the bin size. For example, a 16-binned model considers 81 and 87 to be in the same bin, while a 128-bin prediction considers 87 and 88 as separate bins. We observe that the cross-entropy models train faster to reach their peak performance, especially with coarser granularity like 16-binned labels. While in many cases, 16 bins could be enough to tell if the robot is too loud, it may not help with sensitive listeners.

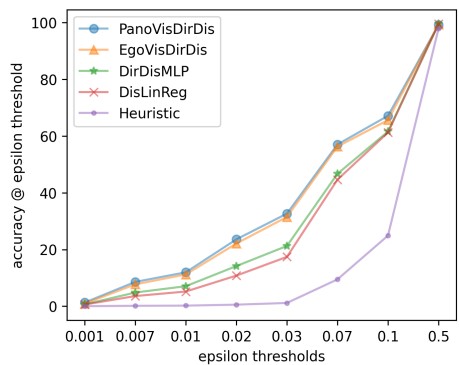

Figure 5: $\epsilon-$threshold accuracy for Heuristic, DisLinReg, DirDisMLP, EgoVisDirDis, and PanoVisDirDis

(ii) *Visual input: Ego-centric vs Panoramic view.* For ego-centric images, we extract the 90°FoV from the panoramic image at the robot's location in the direction of the receiver. We use ResNet-18 as the visual encoder for RGB features. We encode the distance and direction with a linear projection to a 16-dim vector. In Fig. 6, we observe that the learned distributions with ego-view miss outliers with very high values for the longer distances. In Fig. 5, the overall $\epsilon-$accuracy is lower than our Pano- version. While ego-centric is desirable for simplified real-time perception with a camera, it has a limited field of view and lacks the information about architectural geometry in the immediate vicinity of the robot; which can affect how sound travels to the listener.

## 3.3 Analysis

To better understand the model's performance, we create visual representations of two scenarios by mapping the expected max dB value at each location. See Appendix B.3 for visualizations. (i) *Fixed Robot Acoustics Map:* In this scenario, we initialize the agent at a location and compute the model's predictions for different listener locations. The input to the model consists of the same image, with variations in the distance and direction of the listener. This acoustics map helps to predict how many possible listeners in the environment perceive the robot's actions as loud in Sec. 4. (ii) *Fixed Listener Acoustics Map:* In this scenario, we fix the listener's location and compute the max dB heard by the listener when the robot is at different locations in the environment. We construct this acoustic map to estimate the noise cost incurred at each state for planning in Sec. 4.

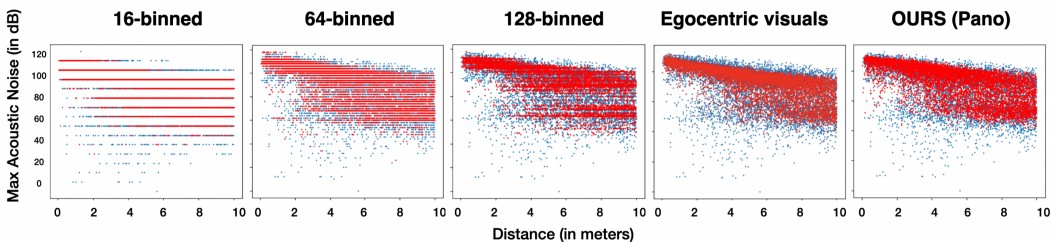

Figure 6: **Ablations**. Data distribution plots for classification models trained with the cross-entropy loss with (a) 16, (b) 64, (c) 128 binned dB values, (d) regression model with an ego-centric view and (e) our proposed model on the right. We observe that the cross-entropy models are easier to train and capture outliers but provide a coarser prediction granularity. The learned distributions with ego-view miss outliers with very high values for the longer distances.

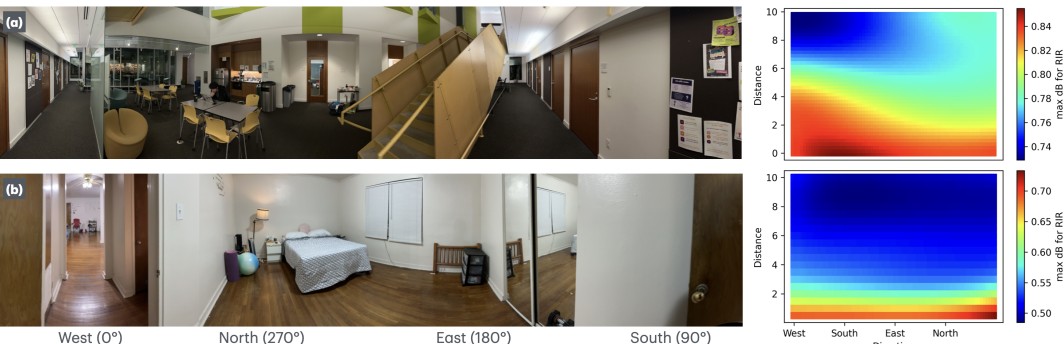

Figure 7: We present our ANP model's predictions for maximum decibels of room impulse response (max dB RIR) with real-world panorama scans. Note that max dB RIR is normalized between 0 to 1. **Top**: a large space in an office building. Here the north faces an open space area, with a corridor from west-to-east direction. There is a staircase close enough in the northeast direction that blocks most of the sound. In the south direction is a wall, and office rooms have closed wooden doors and bulletin boards. **Bottom**: a bedroom of an apartment. The west direction leads to a corridor and then the living area till 9.5m from the robot's location. The bed and a side table are placed in the north, with a wall at about 3.5m. Towards the north are windows and some furniture at about 3 m relative to the robot. In the south direction, a wall and an opened wooden door at 0.5m. The sound measurements are reported in Table 1. Our model's predictions align with that sound decays slowly in the open space with distance and peaks at shorter distances in corridors due to reverberations. The decay is much faster in a smaller room than in an open office space.

## 4 Real world experiments

We propose to evaluate how well the sound loudness is perceived at different listener locations in its environment. Unlike the audio simulators, which have privileged information about the scene mesh, material properties, etc., the agent can only see in its immediate vicinity and knows the relative polar coordinates of the listener. Ideally, we would move the robot to multiple different rooms/apartments and then record the audio. However, practically, robots can be hard to move across apartments and this limits data collection. Thus, an easier method to gather audio responses is to record the sound of robot's actions onto a device (e.g. phone or laptop). Then instead of moving the robot to other locations, we take our device and play back the sound. An added benefit of using recorded audio is that we reduce the variability of the source sound across measurements at different listener locations.

Our experiments use the recorded sounds from a Unitree Go2 and Hello Robot Stretch performing different actions. We use a laptop speaker to play the robot action's audio; acting as the sound source. We use mobile phones to record panoramic images at the robot's location, measure relative distances with Augmented Reality apps, and record audio at the listener location [10].

**How loud will the robot be in my home?** We want to evaluate our model's prediction for the loudness measured at different listener locations from a fixed robot location. Recall in Sec. 3.3;

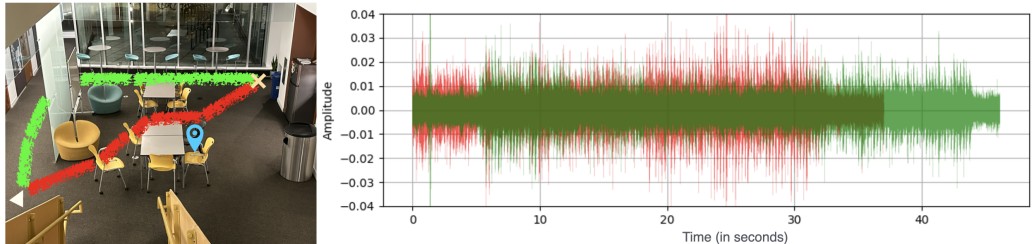

Figure 8: Real-world audio recorded at the listener location (in blue) as the robot starts (from the triangle) and reaches the target (cross). The red path is approximately the shortest distance (32 seconds), but here, the robot goes quite close to the listener. A longer yet quieter path (46 seconds) is shown in green as it navigates behind the separator, thereby diffusing the robot's movement sound that reaches the listener. Best viewed in color.

Table 1: Acoustic measurements of Stretch and Unitree Go2 robot actions sounds, over the distance. The audio recordings were taken in a real Bedroom in an Apartment.

| Distance | Predicted | | Stretch (fast move forward) | | | Unitree Go2 (running) | | |
| Direction | max dB IR | normalized | Real dB | Pred dB | Error | Real dB | Pred dB | Error |
| --- | --- | --- | --- | --- | --- | --- | --- | --- |
| 0m, origin | 120 dB | 1 | 76 | - | | 98 | - | |
| 0.5m S | 87.9 dB | 0.68 | 52 | 51.68 | 0.32 | 70 | 66.64 | 3.36 |
| 1m N | 84.8 dB | 0.66 | 49 | 50.16 | -1.16 | 63 | 64.68 | -1.68 |
| 5m W | 68.5 dB | 0.53 | 47 | 40.28 | 6.72 | 54 | 51.94 | 2.06 |

we discussed acoustic maps for fixed robot and fixed listener locations. We visualize the dense prediction of our model for a few real-world panoramas at a fixed robot location. In Fig.7, we show ANP in (a) a large open office space, and (b) a bedroom. We observe several sim-to-real gaps, as noted in Appendix A; highlighting the distribution shift between sim and real.

Table 1 reports the real decibel sound pressure level (dB SPL) calculated from the audio recorded at different distances in the bedroom environment (as shown in Fig. 7 (b)). We notice that the error is greater at greater distances, and the model underestimates dB specifically in the long corridor leading to the open area in this instance; indicating a wide sim2real gap for audio and vision.

**Can the robot plan for a quieter path?** We want to measure the perceived loudness at a fixed listener location as the robot moves around to reach the target location. A naive solution to reduce robot audio is to increase the distance from the listener. While this heuristic is applicable in open settings, it is most likely sub-optimal in indoor homes. Our main idea with the ANAVI framework is that we can use the ANP to estimate the robot's noise level at the listener and incorporate this as a cost alongside the time to find a path that balances the noise cost and path length. This balance is context dependent, e.g. a person working with or without headphones would change the relative impact of the robot's noise. Using existing LLMs or VLMs with prompting can assign a numeric weight on the noise cost which can be used for the planner. To better understand how the robot's movements are perceived by a listener, we consider the following setting. In Fig 8, we record the panorama at the different locations in the environment. As discussed in Sec. 3.3, we sample a few locations in the environment to record the panorama for acoustic noise predictions at the listener's location. Based on this, we select a quiet path and teleoperate the Unitree Go2 along this path. We also record the shortest path traversed as a control set. For more details, refer to Appendix A.1.

## 5 Related Work

**Acoustics in Learning and embodied AI**. Sound and vibrations have been extensively studied in physics, architectural designs, and acoustic material properties. Recent works in multimodal machine learning have studied the alignment of audio and visual observations [11], acoustic synthesis for novel view [12], and how actions sound [13, 14, 15]. Improved acoustic simulation, with bi-

directional ray tracing techniques [5], has led to realistic room audio simulation and further research in audio-visual matching [16], spatial alignment [17], learning dereverberation [18], audio-goal navigation [19] and audio based learning [20]. [21] present a movable microphone and camera rig to record realistic room impulse response, and show sim2real transfer. We leverage realistic acoustic and visual simulation frameworks to solve an underexplored but practical problem that requires agent to predict acoustic noise loudness at specified location in the environment.

**Acoustics in Robotics**. Existing works [22, 23, 24, 25] consider acoustics as a sensory signal that informs about the robot-object or object-object interactions in the environment. Attending to the audio signal improves the robustness in manipulation tasks; especially where the robot cannot see due to self-occlusion or low-lighting conditions. To simulate human robot interaction, [2] proposed a multisensory platform in a audio-visual simulator similar to ours. In ego-noise prediction task for robots, very few works exist, for example [26] studies self-generated audio imitation. In contrast, our work investigates how the robot can learn to anticipate how the sound generated by its movement at other locations. Additionally, audio is actively studied for easier spatial exploration [27, 28], verbal task specification for improved human-robot interaction [29, 30, 31]. Our work provides tools to further advance the robot's ability to act based on how the audio will be perceived by the listeners.

# 6  Limitations

Acoustics is inherently complex and often noisy (pun unintended), whether in simulation or the real world. In simulation, the holes in mesh reconstructions often reduce the ray tracing efficiency. In the real world, ambient noise and microphone sensitivity usually interfere with reliable and reproducible acoustic measurements. Material absorption, scattering, transmission, and dampening properties are complex, and the resulting reflections and reverberations are hard to measure and predict.

This work uses simulated environments (Matterport3D scans) and audio impulse response simulation (SoundSpaces). Our approach enables controlled experiments and large-scale data collection. However, as discussed in Sec. 4, it may not fully capture the complexities and nuances of real-world environments and acoustic properties. Our current framework does not address the effects of ambient noise. In this work, we focus on first-order principles that directly involve the robot's audio in relation to the audio. However, in real scenarios, the effect of other ambient noises (e.g. a loud TV on) changes how the robot's noise effects the listener. Incorporating other noise sources requires more complex reasoning like sound source separation and noise processing.

Loudness is a subjective and psychological sound pressure; differing based on demographics [32] and prior duration of noise exposure. We use max dB from the impulse response $w(t)$; it is an important yet insufficient aspect to truly measure loudness. In the future, we hope to train models that capture frequency, duration, and other characteristics relevant to perceived sound from room impulse response in simulated and real-world environments.

# 7  Conclusion

We want our future robots to reason about auditory disturbances when operating in human environments like households, hospitals, and offices. Robot assistants in homes and indoor environments generate sound due to their movements or speech through robot's speakers. For any sound produced, robots should be aware of the loudness perceived by the other listeners in the environment.

We present an Acoustic Noise Prediction (ANP) model that uses visual features, along with the listener's distance and directions, to predict the max decibel value of impulse response at the listener. The model enables us to predict the robot's perceived loudness in the environment and to plan routes that lead to less noise perceived by the listener. We show the real-world applicability of our ANAVI framework to mitigate noise to the listeners. With audio noise awareness, we hope that future robot policies can adapt their path, velocity, and speaker's volume to adhere to the environmental noise constraints; thereby becoming well-suited to function in human-inhabited spaces.

**Acknowledgments**

We would like to thank Abitha, Minyoung, Jimin and the CoRL reviewers for useful feedback. This material is based upon work supported by the Defense Advanced Research Projects Agency (DARPA) under Agreement No. HR00112490375.

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

# A Real-world Experiments

## A.1 More details on ANAVI Framework

In Section 4, we provide experiment for robot planning a quieter path. For adapting robot path plans, we detail the ANAVI framework here. See Figure 9.

**(a) Record the environment**: Localize and cache the visual observations. This is similar to the Matterport scans of a physical space that records the localization and panoramic views from each location. Let each location be a node in the graph.

**(b) Run ANP**: Given known (1) location of listeners and (2) the panoramas of the discrete locations of the map: For each node in the graph, use the panorama and the relative polar coordinates of the listener from this node as inputs to the ANP model to predict the max RIR dB. Use the max RIR dB with the robot's action audio files to estimate max dB for each action.

**(c) Time vs. Audio Noise Tradeoff**: Given the social scenario, use LLM prompting or heuristics to decide a weighting factor between the audio noise cost and the time-to-go cost. For example, if a person is having a video call, then the audio noise cost is high with respect to time taken to goal, versus if someone is listening to headphones, then the noise cost is not too high and the robot can optimize for time. With multiple listeners, we can estimate their sound sensitivity, and then incorporate their relative weighted plan into the overall cost.

**(d) Plan**: Use the weighted overall cost with an A* planner (or any other cost informed motion planner) to plan over what actions and path and velocity to take. We discretize slow/normal/fast velocities and encode them as actions with corresponding audio and time costs.

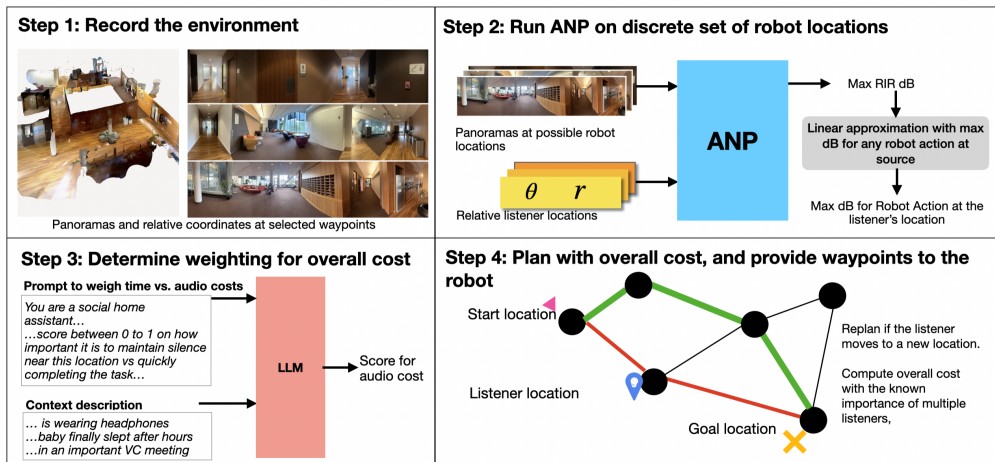

Figure 9: Overview of ANAVI Framework: (a) Scan Environment, (b) Run ANP for listeners, (c) Weigh the audio noise vs. time costs, multiple listeners, and (d) plan with the overall cost.

## A.2 Collecting Real-world Audio

**Setup** We compare the model's prediction with how loud the robot would be in the real world in Section 4. We describe how we collected the data and provide real-world acoustic measurements in Table 2. In this experiment, we fix the robot at a location and vary the listener locations.

We want accessible commodity hardware to ensure easy replicability for researchers so they can use existing resources. We also want to ensure that an easy setup on a mobile robot for future research. To this end, we use mobile phones to capture images, record audio and measure distances.

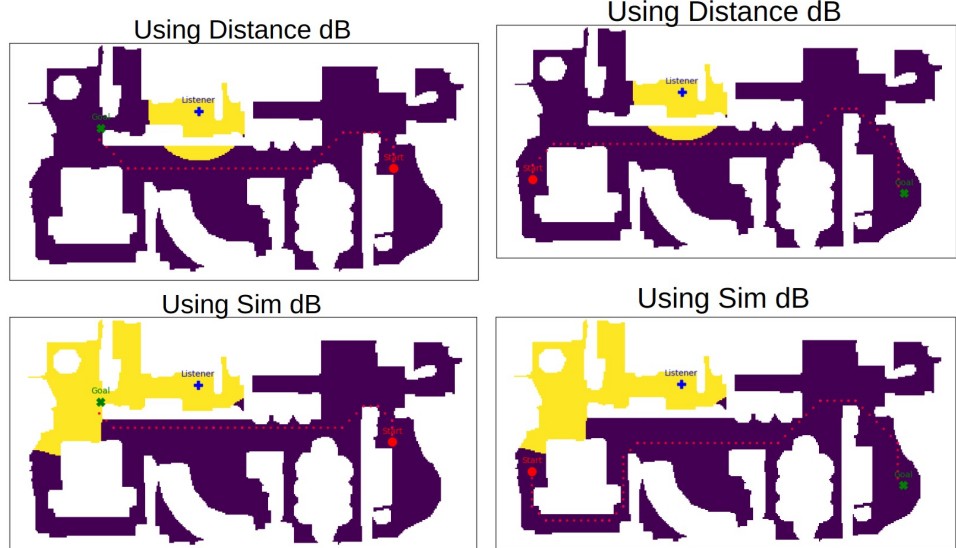

Figure 10: We visualize how accurately estimating the noise cost (using the simulator as opposed to a distance heuristic) can enable better audio-aware paths.

Here we are the steps that we followed after we had recorded our the Stretch and Unitree audio onto our laptop. Our laptop acts as a proxy for the two robots in our experiments, and we place it at the location where we want to pretend the robot is at.

1. Decide an origin (robot location), and take a panorama picture. We select indoor locations like bedrooms, living area, dining area, open and closed working spaces, corridors, stairs, auditoriums and more.

2. Place speaker device with the recorded robot audio at the origin. We used our laptop and set the volume to its maximum. The speaker volume is fixed across all acoustic measurements.

3. Measure distance and direction for a listener location.

4. Place a microphone for recording at the listener location. We used an iPhone for recording.

5. Start the recording on the device and walk back to the origin to play the robot action sound. We note that the walk to the origin and back needs to be trimmed from the audio file so that it only contains the main audio. We thus found it helpful to first say 'Action', then play the audio file for robot action (at max volume) and then say, 'Cut'. After 'Cut' we walked back to the microphone (iPhone in our case) and stopped the recording. Saying 'Action' and 'Cut' helps to trim the audio faster by looking at the audio waveform to trim and only saving the audio in between. Researchers working in pairs can skip this complication and just use hand signals to denote starting and stopping the audio recording and speaker sound.

6. Extract the max decibels from the recorded audio files.

To estimate the max decibels at the origin (sound source), we need to know the volume gain adjustment on the original waveform. We guesstimate it to be a factor of 10, that is 20 dB. Note, the model is likely to have sim-to-real gap and our estimated value also accounts for the systematic error in the model's predictions. To obtain the model's prediction for the max decibel level by each robot's action, we use a linear approximation, that is, multiplying the normalized max dB output with the estimated max decibels at the origin (sound source). We report the error between this value obtained with linear approximation and that from the recorded audio files in the main paper.

**Discussion** Table 2 shows the real audio recorded for Stretch at the fastest forward velocity and Unitree Go2 in running mode using our set-up. We keep the speaker at the origin corresponding to Fig. 11 and vary our listener location. As we can see in the figure, moving west or north stays in

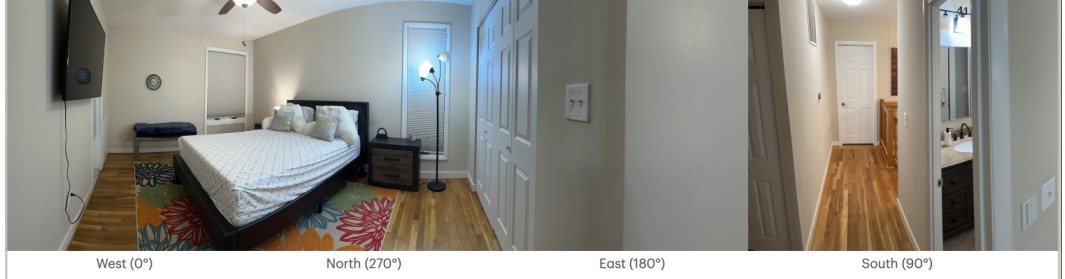

| West (0°) | North (270°) | East (180°) | South (90°) |

Figure 11: Panorama taken from the sound source location. Note the directions west, north are in the room, facing east is a wall and the south direction leads out of the room into the hallway.

Table 2: Real Audio for a Bedroom in a Single Family House.

| Distance-Direction | Line-of-sight | Outside the room | Stretch Real dB | Unitree Go2 Real dB |
|---|---|---|---|---|
| 1m-west | yes | no | 54.2 | 69.7 |
| 1m-south | yes | yes | 52.1 | 67.1 |
| 1m-south-1m-east | no | yes | 49.7 | 66.1 |
| 2m-north | yes | no | 51.7 | 67.3 |
| 2m-south | yes | yes | 50.5 | 66.7 |
| 2m-north-2m-west | no | no | 45.3 | 61.1 |
| 3m-west | yes | no | 48.6 | 65.4 |
| 3m-south | yes | yes | 48.2 | 65.4 |

the room while moving south or east corresponds to moving outside the room. We first compare the dB between "1m-west" and "1m-south" and observe how moving the sensor out the room decreases dB (as expected). We see this difference consistently although with smaller values as the distance from the source increases. This highlights how architectural geometries can have non-trivial impacts on audio (as expected). Additionally, we observe how line-of-sight between the source and listener affects the dB values. In case of obstacles like the bed or wall, the dB values decrease faster as compared to longer distances with line-of-sight (e.g. "1m-south-1m-east" is lower than "2m-north").

### A.3 Qualitative Evaluations on Real-World Panoramas

We collect real-world panorama of indoor environments to visualize the model's predictions. We focus on sim-to-real performance of the audio prediction, as once accurately estimated, these values can be weighted against distance for audio-informed planners.

**Setup** To collect real world panoramas, we requested graduate students to contribute these panoramic images, by capturing their surrounding indoor environment for acoustic profiling. Contributors used their mobile phones at zoom level 1x and took a single panoramic image which we then resized to 256×1024. The images are then fed into a neural network that predicts the maximum decibel level at a given distance and direction from the robot. The ANP neural network is trained on a dataset of simulated Matterport renderings, and we qualitatively evaluate its performance on a dataset of real-world panoramas. Below we use Figure 12 to explain our visualizations.

The first row shows the real-world panorama. The leftmost and rightmost edges correspond to 45°, and we change angles in clockwise direction from left-to-right. The reason for non-increasing order of angles on x-axis is that panorama's are taken left-to-right, that is, clockwise, whereas angles are measured anti-clockwise. Note that the cardinal directions indicated on the plots does not imply that the panoramo starting direction and are only for illustration purposes.

The second row shows heatmap plot with the model's prediction for different distance and direction values. The direction is shown on x-axis, covering 360°, and the distances on y-axis range from 0 to

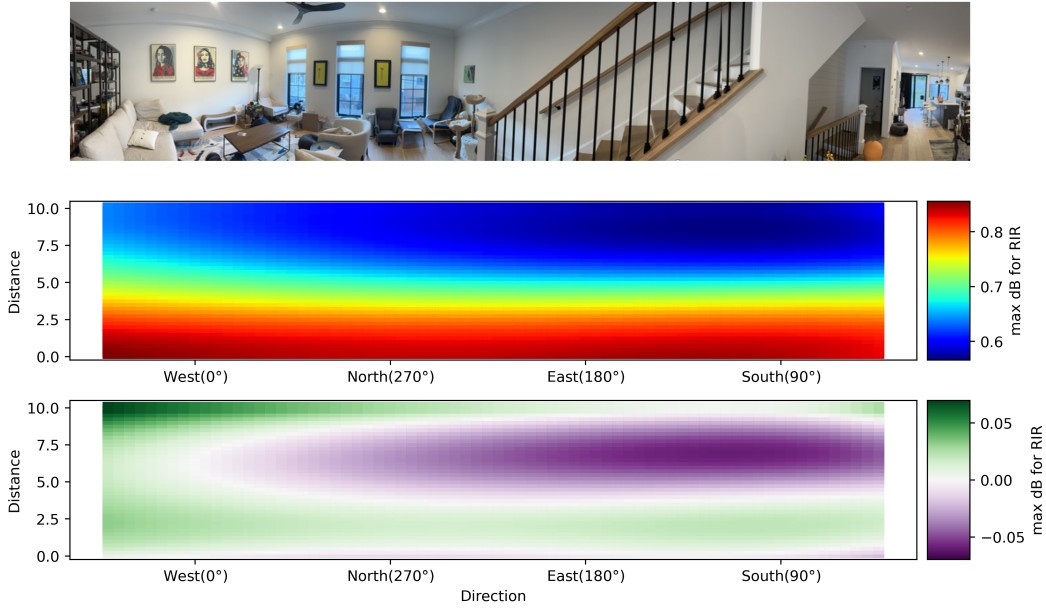

Figure 12: **Setup for ANP Predictions on Real World Images**. This image visualizes ANP model predictions similar to Figure 7 from the main paper. The top row depicts a real-world panaromic image. The second row plots the maximum decibels of room impulse response $\hat{y}_{\text{ANP}}$ (max dB RIR). Note again that max dB RIR is normalized between 0 to 1. The third row plots the difference between the model's prediction and the linear regression prediction, i.e. $\hat{y}_{\text{ANP}} - \hat{y}_{linreg}$.

10 meters. Note that the direction is aligned with the visual image, i.e. it starts from -45° to 0, then reduces 270°, 180° 90°, and finally to 45°.

In the third row, we plot the difference from the distance based baseline. Since the distance based baseline doesn't depend on the image, we visualize the variation in our model's prediction from the baseline. Here the green indicates where the model predicts significantly higher normalized max dB values than distance-based linear regression model, and purple indicates significantly lower normalized max dB values than naive baseline. Note that the dB values are normalized from 0 to 1.

**Discussion** We see that the model mostly predicts that the max dB values monotonically decrease with distance, though it may vary with direction. This is an important initial sign of life, as image features are very high-dimensional as compared to a scalar number used for relative distance. Nevertheless, the model largely learns to attend to the distance as input.

*Does our model understand acoustic features beyond distance?* In Figure 13, we show cherry-picked examples of (13a) an open office area , (13b) a bedroom in a single family house. In Fig. 13a, we observe that the model predicts high dB values at larger distances in open spaces, as in north-east direction. In terms of the difference with the baseline, we observe that the model predicts higher sound intensity than the baseline at larger distances. In Fig. 13b, we observe that the model predicts low values at larger distance for some areas where there is a wall. Walls acts as an obstacle for sound intensity (see east direction). Additionally, we observe that the model predicts higher values at larger distances in open spaces, within the room (see west and north directions), than in the outside corridor (see south direction). In the second subplot, we note that our model predicts higher max dB RIR than the baseline at larger distances, and lower values near the sound source.

Through the qualitative real-world data evaluation, we note several sim-to-real gaps. First, the real images are higher quality than the Habitat environment's Matterport3D renderings. This includes differences in how visually environment attributes as inferred, such as depth, material texture, and architectural geometry. Second, lighting changes and variations are more drastic in real-world than the static Matterport renderings in simulation. This has implications on acoustic noise prediction

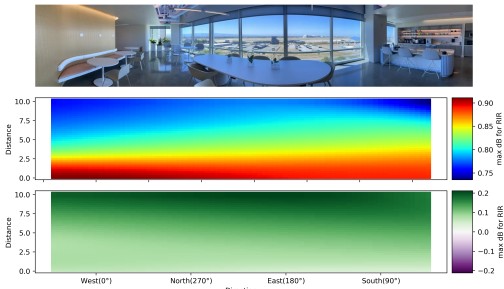

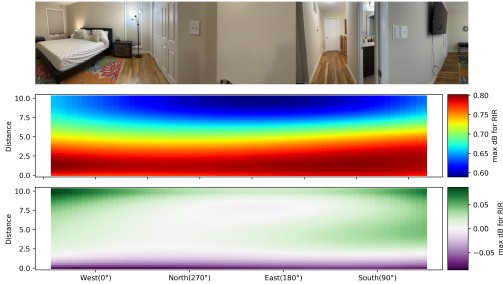

(a) **Open office area**. High dB values predicted at larger distances for open spaces (see east), and at shorter distances in the corners, near walls (see west and south-west). ANP predicts higher sound intensity than the baseline, especially at larger distances, as shown in green.

(b) **Bedroom in a single family house** Low dB values predicts near wall as an obstacle (see east) and outside the room (see south). High dB values at larger distances in open spaces (see west). ANP predicts higher values than the baseline at larger distances (green), and lower values near source (purple).

Figure 13: We show cherry-picked examples of model's prediction in (a) an open office area, and (b) a bedroom in a single family house.

models deployed on the home robots, and we need to improve the robustness to lighting variations for the same scene. Third, the height perspective of our images and the relative angle to the ground differed between contributors, while this was a fixed constant in simulation. Differences in camera height, pan and tilt requires slightly different interpretation of the distances from visual features.

We observe that the current model performs poorly on these diverse set of real-world panoramic images. The first row (Figures 14a and 14b) shows two panoramas of corridors where we expect corridors to have high predicted db regardless of sensor distance while the corresponding wall area have small values when the distance is large (as then the sensor would be behind the wall). Instead, we did not see any noticeable correlation to the corridors. Upon closer inspection on predictions, the model seems to struggle with identifying walls and their impact on sound. We expect high dB predictions in rooms when the distance falls in the room, but then a large drop-off to low dB prediction outside rooms. But in Fig 14c, 14e and 14g, we do not see the drop-off.

Our results show that zero-shot generalization to real panoramas is not easy. It is unclear if this issue lies in simulation inaccuracies, an inadequate ANP model, or the distribution shift to real-world panoramas. Future work should address these sim-to-real differences to enable using an ANP model effectively in real environments. In future, we hope to bridge the sim2real gap with (1) effective domain randomization for camera poses, lighting and robot's viewing angles, (2) improving model training with auxiliary losses, and (3) fine-tuning model with real-world measurements for acoustics, distances and visuals, collected from diverse indoor environments.

# B  Simulation

## B.1  Training Data Generation

We generate training data for ANP in simulation using SoundSpaces 2.0. Here we provide additional details for the data generation process: (i) *Uniform sampling across the map.* We first divide the map into grids to get 100 points. These points serve as circle centers to sample a suitable navigable point within 20 meters radius for the robot's location as the source. We then use the robot's location as the circle center and sample a listener location within 10 meters radius. (ii) *Visual panorama for the robot's source location.* We sample RGB camera observations 4 times with rotations of 90 degrees at $256 \times 256$ resolution and 90 degrees field-of-view. We discuss the audio data generation below as part of room impulse response (RIR).

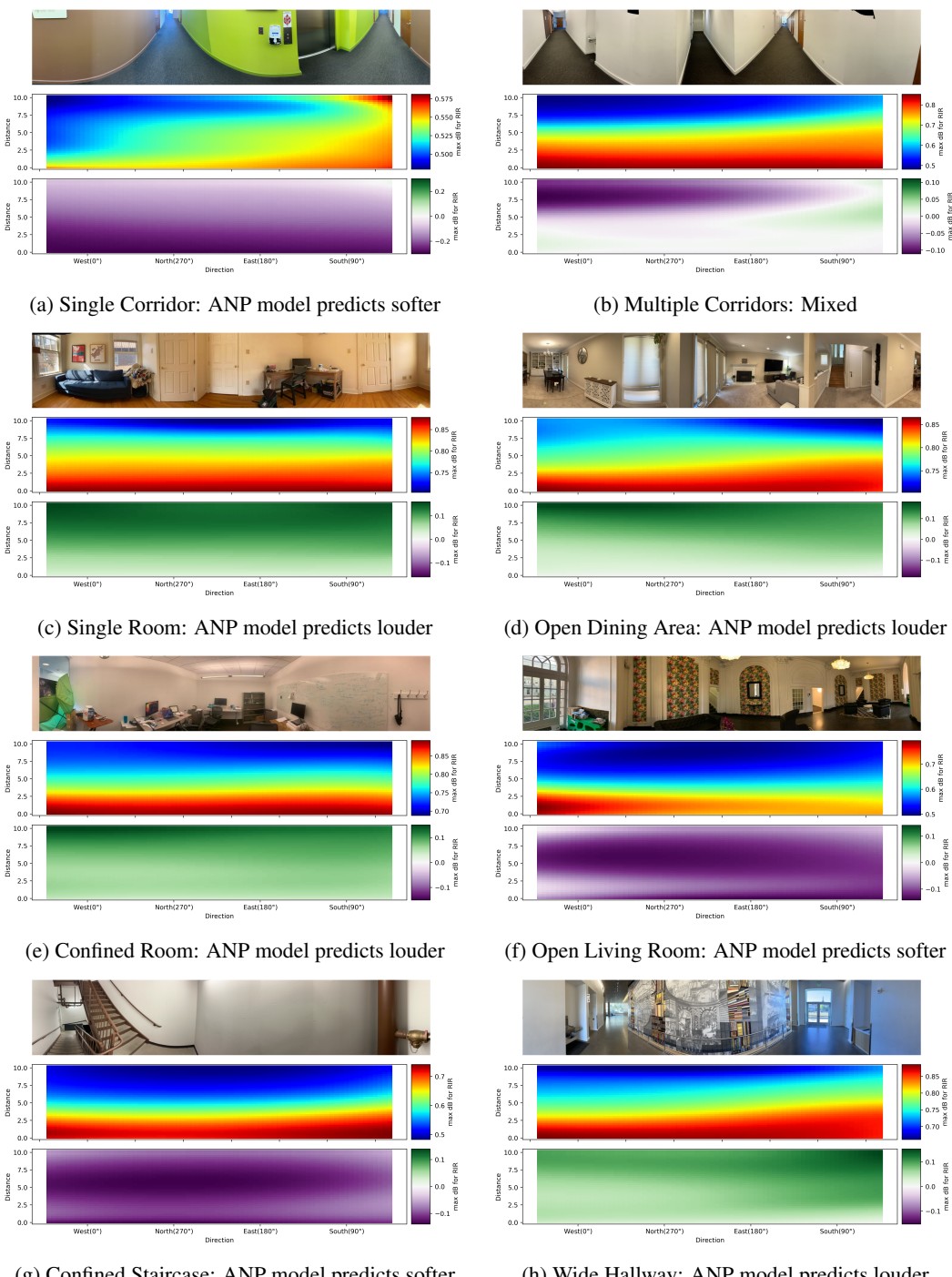

(a) Single Corridor: ANP model predicts softer

(b) Multiple Corridors: Mixed

(c) Single Room: ANP model predicts louder

(d) Open Dining Area: ANP model predicts louder

(e) Confined Room: ANP model predicts louder

(f) Open Living Room: ANP model predicts softer

(g) Confined Staircase: ANP model predicts softer

(h) Wide Hallway: ANP model predicts louder

Figure 14: We see a severe sim-to-real gap when applying the ANP model to real world panoramas. For each plot, the top row shows a real world panorama, second row the ANP predictions depending on the distance and angle to the sensor location, and the third row the difference compared a basic linear regression estimate (green = ANP predicts louder). Unfortunately, the model is unable to consistently capture meaningful visual features like walls or corridors that would effect dB.

## B.2 More details on Room Impulse Response (RIR)

An impulse response is the output signal (sound) that results when an audio system or environment is excited by an idealized impulse signal at a specific location. An impulse signal has a very short duration and theoretically contains all frequencies.

We notionally define the simulated impulse response at the listener's location as $w(t)$, a time-domain waveform. We obtain an impulse response using standard techniques that we define below. Let an impulse be generated at the robot's location $s = \{s_x, s_y\}$ and heard by a listener at location $l = \{l_x, l_y\}$. Then, using bi-directional ray tracing in simulation, we obtain the maximum sound intensity at the listener location. First, we convert the simulated IR waveform $w(t)$ from the time domain to the frequency domain $W(f)$ with Fast Fourier transform. Second, we compute the sound intensity $I(f) = W(f)^2/(\rho * c)$, where $\rho$ is air density and $c$ is the speed of sound in air.

We extract the maximum sound intensity as $I_{max} = \max I(f)$. Since the faintest sound human ears can hear is considered $I_o = 10^{-12} W/m^2$, the sound intensity is converted into to decibels by $dB_{max} = 10 \log_{10} I_{max} + 120$. As we make a modeling design choice to assume an impulse of $1 W/m^2$ in simulation, we define 120 dB sound pressure level at the source. A large impulse value ensures a good signal-to-noise ratio while avoiding distortion. To ensure that the decibel values are normalized for training, we assume the highest decibel value as 128 and normalize it between 0 and 1 to get target labels $y$. For simplicity, we assume that the listener captures a mono-channel audio.

More concretely, we do the following steps:
$$w(t) = \text{SimulateIR}(\text{from} = p_{source}, \text{at} = p_{receiver}) \tag{1}$$
$$W(f) = \text{Fast Fourier Transform}(w(t)) \tag{2}$$
$$I(f) = (W(f)^2)/(\rho * c) \tag{3}$$
$$I_{max} = \max I(f) \tag{4}$$
$$I_{max} = \text{clip}(I_{max}, \min = 10^{-12}, \max = 10^{0.8}) \tag{5}$$
$$dB_{max} = 10 \log_{10} I_{max} + 120 \tag{6}$$
$$y = dB_{max}/128 \tag{7}$$

Here $w(t)$ is the time-domain waveform generated at the receiver's location, $W(f)$ is the frequency domain waveform, $I(f)$ is the frequency domain sound intensity through air computed by root mean square. The $\rho$ is air density and $c$ is speed of sound in air. We use the maximum sound intensity and convert it to decibels. To ensure that the decibel values are normalized for training, we assume the highest value of decibel as 128 and compute target labels $y$. Since the faintest sound human ears can hear is considered $I_o = 10^{-12} W/m^2$, we convert the max sound intensity into to decibels by $dB_{max} = 10 \log_{10}(I_{max}/I_o) = 10 \log_{10} I_{max} + 120$.

## B.3 Visualization of Fixed Robot and Fixed Listener maps

In Section 3.3, we discuss the fixed listener and fixed robot prediction maps for analysis. Figure 15 shows a few of the fixed listener and robot maps, and highlights the potential failure cases of the ANP model.

# C Discussion and Broader Scope

With audio-level prediction, we are one step closer to future home robots. Nevertheless, we must develop reliable and calibrated loudness-aware text-to-speech response and navigation policies. Current acoustic map predictions must be calibrated to real-world audio and appropriate weighting relative to time cost. Learning from visual features is prone to overfitting. Collecting audio-visual measurement data over a larger, diverse set of environments can help alleviate this. This will eventually facilitate context-aware robot policies for different loudness sensitivities.

Our current framework does not address the effects of ambient noise. In this work, we focus on first-order principles that directly involve the robot's audio in relation to the audio. However, in

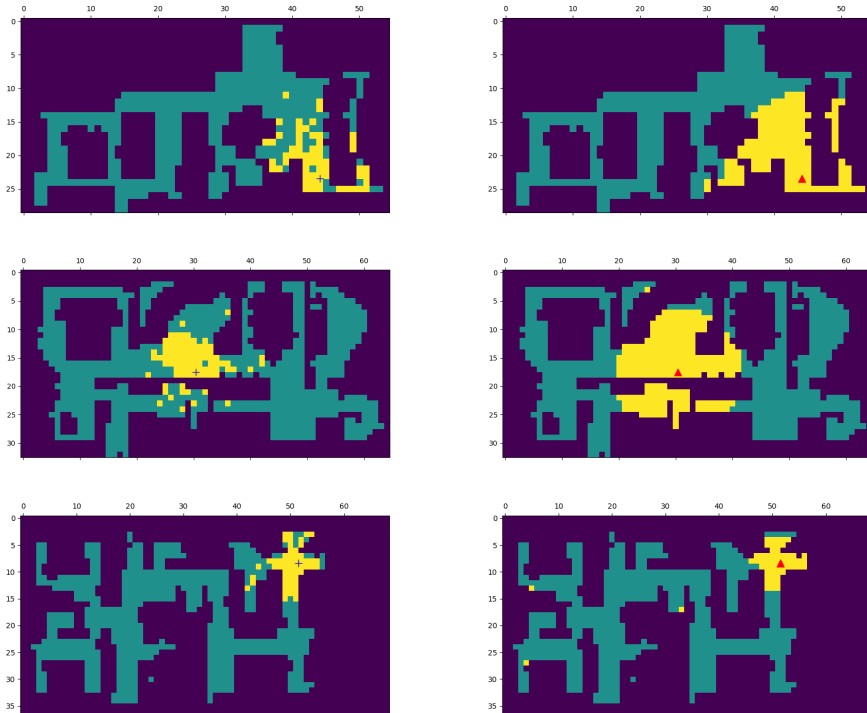

Figure 15: ANP Predictions over the BEV map for (Left) Fixed listener (indicated with blue plus sign) and (right) Fixed robot (indicated with red triangle). Note that ANP uses panorama from robot's point of view, implying that the fixed listener maps use different visual features whereas fixed robot uses the same visual for each cell. The relative distance and direction from robot to the listener changes for each cell. We show all locations with 96 dB (more than 85%) in yellow, rest in blue and non-navigable regions at height 0 in purple. **Top**: The fixed listener predictions are spread more along the corridor (up-down) than (left-right). The fixed robot plot is more smoothly and uniformly spread across the map. **Middle**: The fixed listener predictions align with intuition that the sound should be louder in the open space compared to the area at same distance but separated by a wall. The fixed robot predictions plot follows the distance heuristic but not the common intuition. **Bottom**: Both fixed listener and fixed robot maps show the expected spread of sound at the cross-intersection.

real scenarios, the effect of other ambient noises (e.g. a loud TV on) changes how the robot's noise effects the listener. Incorporating other noise sources requires more complex reasoning like sound source separation and noise processing that is less robotics related. For example, this could involve predicting that a refrigerator makes ambient noise and thus the robot can be louder near it, but this has been studied in acoustic literature and was not the main focus of our work. Future work could extend ANAVI framework to handle this in two ways: (i) by adjusting the "Time vs Audio Noise Tradeoff" (See A.1 (c)) based on the prompt (e.g. the TV is on) that LLM can use to adjust the audio noise tradeoff. (ii) by modifying ANP to include ambient noise as 'signal' and robot movement sounds as 'noise' to guide the robot's path and velocity planning.

