# OpenReview forum: "ANAVI: Audio Noise Awareness using Visual of Indoor environments for NAVIgation"
_robot-learning.org/CoRL/2024/Conference — CoRL 2024_

### Official Review · Reviewer_WfSy · 2024-07-16
**Interesting motivation, but plenty of room for improvement for execution**

**Originality:** 2
**Technical Quality:** 3
**Clarity Of Presentation:** 3
**Potential Impact:** 1
**Recommendation:** 1
**Confidence:** 4

**Review:**

The paper has a few noteworthy strengths. First of all, the problem domain seems interesting and under-studied: there hasn’t been much literature on robot’s understanding of how its own noise will impact other users in the same environment. The simulation setup as well as the data collection process seem legitimate. The intuition that panorama images should help the prediction also makes a lot of sense. It’s also appreciated that the authors take the time to conduct sim2real transfer experiments with real-world audios and visuals.

However, the paper is not without room for improvement. First of all, the scope of the paper seems limited. It’s unclear what “visual interaction” means in the title. Initially, I would expect the paper to study how robots might cause noise (e.g. impact sound as the robot manipulates objects in the scene), but the paper only studies the robot noise during navigation.

While the authors did a decent job of comparing different baselines and ablated models, all the models (including the proposed method) seem overly simplistic and lack technical novelty. There are in fact many potential for innovation, for example, is resnet-18 the best visual encoder for processing panoramic images? Can the robot also hear its own noise, and use that as part of the observation as well (this will amount to multi-modal learning).

In terms of experimental evaluation, the results are a bit lacking. Figure 4 seems to be the only quantitative results for the sim experiments and it’s a bit difficult to read (dots are overlapping). For the real-world experiments, the authors observed some sim2real gap, but didn’t manage to carefully study these gaps and provide appropriate analysis.

The relevance to robotics is slightly questionable as well, since the only real robot experiments are collecting real-audio with robots sitting idle, without any robot control or decision making. For instance, it might be interesting to train a robot policy that learns the tradeoff between quieter path and shorter path, based on some user’s preference (e.g. maybe the user is okay with louder paths during the day, but not okay during the night)

Finally, some important related works are not properly cited, such as Sonicverse, which also trains a model (in this case, audio visual navigation), and performs sim2real transfer.

**Quality Of The Limitations Section:**

3

**Questions For Rebuttal:**

- Have you considered including audio input for your ANP model (i.e. robot hearing its own noise)? It would be interesting to find a good way to fuse the two modalities.
- Is there any chance you can include a table similar to Table 1 for the sim experiments?
- Where is the main source of sim2real gap? Is it because the model is susceptible to the sim2real gap for the panoramic images? Or is it because it’s trained on unrealistic data i.e. SoundSpaces 2.0 is not accurate enough?

**Robotics Focus:**

2

**Summary Of Paper:**

The paper proposes Audio Noise Awareness by Visual Interaction (ANAVI) for robots to predict the maximum decibel of the Room Impulse Response at the listener’s location, given the listener’s direction, distance, as well as the RGB panorama image of the robots. The authors collect the dataset in SoundSpaces 2.0 and train a few baselines and ablations. The authors demonstrated superior performance for the proposed method, and also tested the model in the real world, i.e. collecting real-world audio and compare against model predictions based on real-world images.

**Summary Of Recommendation:**

I would recommend strong reject since the task setup seems relatively simple, has limited implication / impact for robotics, and the proposed method lacks algorithmic novelty.

---

### Official Review · Reviewer_XYtK · 2024-07-20
**Original and interesting problem for robots in real home environments**

**Originality:** 4
**Technical Quality:** 4
**Clarity Of Presentation:** 1
**Potential Impact:** 3
**Recommendation:** 4
**Confidence:** 4

**Review:**

The paper is original and interesting. Such a problem is seldom addresed in the current literature. However the first part of the paper is very difficult to read with awkward sentences in the abstract and introduction sections. The following sections get fortunately better. The authors are strongly suggested to proofread their paper. The experiments on robots, as well as the navigation planning with noise-constraint are valuable additions,

1. The reviewer does not understand the use of RIR. Soundspace can simulate any sound. RIR is good to infer the acoustic properties of a room. However the robot noise is far from RIR. It would be closer to white noise.

2. Relate to the previous question, the authors chose simplified assumption about the robot noise. The reviewer believes that this noise depends on the proprioception/ mechanical structure of each robot. How would the authors take this into consideration for anavi. A quieter and louder robot would be treated the same in the current network.


3. Comments, the disturbance of the robot noise could also be expressed as a SNR. This would be a better metric to quantify the relative disturbance instead of the max dB.

4. line 251, the reviewer disagrees with this claims. There is a vast literature in Robot audition that address tasks such a sound localization, separation or navigation beyond the robot-object interaction. The authors are either encourage to extend the related work section or change the title to restrict the scope. Acoustics in Robotics is too broad.



Edit: Given the clarification provided by the authors, I recommend the paper acceptation.

**Quality Of The Limitations Section:**

2

**Questions For Rebuttal:**

The questions and issues are listed above.

**Robotics Focus:**

4

**Summary Of Paper:**

This paper introduces anavi, a neural network predicting the level of noise of a robot in real environment. Based on a panaramic view, and a potential listener position in the scene, the network predict the sound level from a RIR. The netwrok is trained as a regression model with groundtruth sound level provided by Sound space.

**Summary Of Recommendation:**

Substantial proof reading and some clarifications are needed

---

### Official Review · Reviewer_i2im · 2024-07-31
**Relevant work but needs minor improvements**

**Originality:** 3
**Technical Quality:** 4
**Clarity Of Presentation:** 3
**Potential Impact:** 3
**Recommendation:** 3
**Confidence:** 3

**Review:**

The paper presents ANAVI, a framework designed to enhance robots' awareness of the noise they produce during their actions in indoor environments. By integrating visual features with audio predictions, the ANAVI system aims to allow robots to adapt their behavior to minimize noise disturbance. The authors have developed an Acoustic Noise Prediction (ANP) model that uses visual features to predict the decibel levels of sounds produced by the robot at various listener locations. The paper includes detailed experiments conducted both in simulation and real-world environments to validate the model's effectiveness.

- The paper is generally well-structured and flows logically. However, the introduction could benefit from a clearer definition of the problem and a more concise statement of the paper's contributions.
- The architecture and functioning of the ANP model are well-explained, but adding a more detailed discussion on why specific design choices were made (e.g., the use of ResNet-18) would enhance understanding.
- The use of ϵ-thresholded accuracy as a metric is innovative. However, a more in-depth comparison with other standard metrics in noise prediction could strengthen the evaluation section.
- The real-world experiments are a significant strength of this paper. Including more scenarios and discussing the challenges faced during these experiments in greater detail would provide a more comprehensive validation of the model.

Overall, the paper presents a novel and practical approach to improving robots' noise awareness using visual features. The combination of simulation and real-world experiments adds robustness to the findings. However, the paper would benefit from additional clarity in problem definition, more detailed explanation of design choices, and a broader evaluation of the model's generalization capabilities.



Minor comments:
Page 1, Line 4: "ok" should be "okay."
Page 3, Line 73: "bi-directional" should be "bidirectional."
Page 8, Line 258: "agent to predict" should be "agent predicting."
Some figures (e.g., Figure 1) could be larger and clearer with labels for each point/arrow. The labels and legends should be more readable.
Table 1: The units for the error should be consistent (either in decibels or a percentage).

**Quality Of The Limitations Section:**

2

**Questions For Rebuttal:**

- How well does the ANP model generalize to completely new environments not seen during training? Have the authors tested the model's performance in such scenarios?
- How does the model account for varying levels of ambient noise that might affect the listener's perception in real-world settings?
- Can the authors provide more details on why they chose ϵ = 0.007 as a reasonable threshold? How does this threshold compare to human perception differences in real-world scenarios?

**Robotics Focus:**

4

**Summary Of Paper:**

By integrating visual features with audio predictions, the ANAVI system aims to allow robots to adapt their behavior to minimize noise disturbance.

**Summary Of Recommendation:**

This paper makes a significant contribution to the field of robot learning and should be accepted after addressing the minor comments and clarifying a few key aspects of the model and its evaluation.

---

### Official Review · Reviewer_2dvx · 2024-08-01
**Model for noise level prediction based on images with potential applications in robotics.**

**Originality:** 2
**Technical Quality:** 3
**Clarity Of Presentation:** 3
**Potential Impact:** 2
**Recommendation:** 3
**Confidence:** 4

**Review:**

The ability to anticipate how the noise produced by an agent is perceived by others is a crucial component of social intelligence and is highly relevant to HRI. The approach of using simulation with recorded 3D scenes to simulate sound to generate training data has significant potential.

- the paper is well written and easy to read.
- the work on the part of data collection and model training seems to be coherent and is convincing
- the learned model shows significantly better results in comparison to the baseline

## General Remarks

- the larger structure and the title of the paper seem somewhat misleading:
the paper begins with a large scope proposing a general framework "Audio Noise Awareness by Visual Interaction" (ANAVI), however
ANAVI is only mentioned four times only in the first page: (1) in the title, (2) in the abstract, (3) line 36, (4) footnote 1 at the page 1.
The definition (line 36) is very general, and the remainder of the paper focuses on the ANP.
The concrete definition of the framework ANAVI remains unclear.
A clearer definition of the ANAVI framework and a discussion of how the proposed ANP model relates to it is needed.

- the proposed approach is not specific to robotics and does not make use of specific capabilities of the robots,
such as having a mental map of the environment and modulate its actions to move more quietly.

- the paper considers only generalized sound impulse response
  - different actions of the robot produce sounds that are perceived differently by an observer, some actions might be more disturbing than others
  - line 52: sound is also produced by the interaction with the environment, e.g., loud steps on the wood floor
  - quieter path can also result from choosing to walk over a carpet instead of the wood floor, or making steps more carefully, i.e., modifying the actions

- the real-world experiments were conducted with a laptop emitting the robot noise and iPhone to record panoramic images and to simulate a listener.
  - this setup come with limitations that should be addressed (e.g., the sound emitted from the laptop is directional in contrast to a robot walking over the floor)

- potentially relevant work on ego-noise prediction:
  - Pico, A., Schillaci, G., Hafner, V.V., Lara, B. (2016), How Do I Sound Like? Forward Models for Robot Ego-Noise Prediction, Proceedings of the 6th Joint IEEE International Conference on Development and Learning and on Epigenetic Robotics (ICDL-EpiRob), pp. 246-251, Paris, France. DOI 10.1109/DEVLRN.2016.7846826

## Concrete Remarks

- Figure 4:
  - why is "Heuristic" missing in this plot?
  - the dots overlap and are hard to distinguish. Maybe use lines between the dots to visualize the progression (line plot with dots).

- Figure 8: right plot is difficult to interpret and hard to distinguish in black and white prints. Maybe a simpler plot visualizing the loudness would be better.

- 3.3 Analysis: the section mentions "Fixed Robot Acoustics Map" and the "Fixed Listener Acoustics Map" that can be found in the supplementary material.
I was not able to find the visualizations of the maps. Perhaps pointing to a specific place in the supplementary material would be helpful.

**Quality Of The Limitations Section:**

2

**Questions For Rebuttal:**

- does the simulation contain material properties, e.g., simulating different sound absorption or reflection properties?

- is it planned to expand the simulation to physically simulating a robot and its sounds?

**Robotics Focus:**

2

**Summary Of Paper:**

The paper is aiming at enabling a robot to learn about the noise resulting from its actions, and to adapt its actions in such a way as to reduce the unintended noise that could disturb humans. For this, the paper proposes a model - "Acoustic Noise Predictor" (ANP) - predicting the expected perceived noise level at a given relative location from the source, based on visual information (panoramic images). The model is trained and evaluated in simulation with real 3D scans. A comparison with baseline method shows better performance of the proposed model. Basic experiments in a simplified real-world setup provide a proof of concept. A laptop emitting robot noise, and an iPhone providing images and acting as a listener were used.

**Summary Of Recommendation:**

The proposed approach provides incremental novelty by applying existing methods and data from other domains in a scenario with has potential application in robotics.

---

### Author Rebuttal · Authors · 2024-08-13

Please find attached the full paper and supplementary draft with suggested changes in this zip folder.
Additionally, we include the updated audio recordings of Unitree go2 for for Figure 8, that are for (i) the fastest path in running mode in 11 seconds, and (ii) a quieter path in walking mode in 15 seconds.

**Update**: As suggested by reviewers i2im and XYtK, we have updated our limitations section about ambient noise incorporation and listed potential future directions.

---

### Decision · Program_Chairs · 2024-09-04

**Decision:**

Accept

**Comment:**

This paper presents a learning-based approach for predicting acoustic noise with the goal of allowing a robot to adapt its behaviour to minimise noise disturbance, particularly in an assistive settings. Reviewers note that this problem is highly relevant and less studied in robotics. However, there are concerns regarding generalisation to new environments and effect of ambient disturbances. As has come up during the rebuttal phase, the paper in the present form needs a major revision in terms of clarity on the technical approach and results. A revised version of this submission has the potential to be impactful in the robotics community.